# Quasi-Zero Stiffness Vibration Sensing and Energy Harvesting Integration Based on Buckled Piezoelectric Euler Beam

**DOI:** 10.3390/s24010153

**Published:** 2023-12-27

**Authors:** Jiying Tuo, Xiaonan Xu, Jun Li, Tianlang Dai, Zilin Liu

**Affiliations:** 1Key Laboratory of Advanced Manufacture Technology for Automobile Parts, Chongqing University of Technology, Ministry of Education, Chongqing 400054, China; tjy@cqut.edu.cn (J.T.); xiaoandmeng99@stu.cqut.edu.cn (X.X.); 52230406118@stu.cqut.edu.cn (J.L.); 52220406166@stu.cqut.edu.cn (T.D.); 2Chongqing Tsingshan Industrial Co., Ltd., Chongqing 402761, China

**Keywords:** vibration sensor, energy harvesting, quasi-zero stiffness, buckled piezoelectric Euler beam, absolute motion

## Abstract

This paper presents a novel quasi-zero stiffness vibration sensing and energy harvesting integration system for absolute displacement measurements based on a buckled piezoelectric Euler beam (BPEB) with quasi-zero stiffness (QZS) characteristics. On one hand, BPEB provides negative stiffness to the system, thus creating a vibration-free point within the system and transforming the absolute displacement measurement problem into a relative motion sensing problem. On the other hand, during the measurement process, the BPEB collects the vibration energy from the system, which can provide electrical energy for low-power relative motion sensing devices and remarkably suppress the frequency range of the jump phenomenon, thereby further expanding the frequency domain measurement range of the sensing system. The research results have shown that this system can measure the absolute motion signal of the tested object in low-frequency vibration with small excitation. By adjusting parameters such as the force–electric coupling coefficient and damping ratio, the measurement accuracy of the sensing system can be improved. Furthermore, the system can convert the mechanical energy of vibrations into electrical energy to power the surrounding low-power sensors or provide partial power. This could potentially achieve self-powering integrated quasi-zero stiffness vibration sensing, offering another approach and possibility for the automation development in wireless sensing systems and the Internet of Things field.

## 1. Introduction

There is a widespread demand for displacement measurement in people’s production and daily lives, such as isolation for moving vehicles [1,2], ship-mounted optical instrument protection [3], precision engineering [4], and scientific research [5]. Particularly in low-frequency vibration applications, it is often necessary to utilize the displacement caused by vibration to better describe system motion or achieve effective wideband vibration isolation. Accelerometers are widely used to measure the absolute motion of a dynamic system. However, it is almost impossible to accurately retrieve the absolute displacement from the measured acceleration signal in real time [6,7]. Geophone is commonly applied to provide absolute velocity measurements of motion. However, these types of sensors are often bulky and relatively expensive [3]. A wide range of methods, including laser displacement sensors [3] and Hall displacement sensors [3], can be used for the accurate measurement of relative displacement. However, these methods require a stationary reference point during usage.

In recent years, researchers have proposed various methods to improve displacement measurement techniques, including the use of new sensing materials, nano-fabrication technology, quasi-zero stiffness structures, and more. Over the past three decades, these structures have been extensively and deeply studied primarily as high-performance low-frequency vibration isolators. By significantly reducing the system resonance peak and its frequency, an absolute stable point in a broadband frequency domain can be created in the QZS system, and it is used in broadband vibration displacement measurement. By using two pre-deformed horizontal springs as the negative stiffness corrector, Sun et al. [6,7] presented measurement methods for mobile platforms. By employing a pre-deformed scissor-like structure, Jing et al. [7] proposed a 3-D QZS-based vibration sensor system for 3-D absolute motion measurement. In 1958, Molyneux [8] first designed a passive vibration isolation device consisting of two horizontal springs and one vertical spring. The high-static low-dynamic (HSLD) characteristic of the device can be achieved by changing the geometric structure of the elastic elements. Kamil Kocak et al. [9] employed flexible beams and a quasi-zero stiffness structure to reduce the starting isolation frequency range. Chen et al. [10] used a combination of quasi-zero stiffness isolation systems to significantly suppress the vibration of a large vehicle-mounted optoelectronic tracking device under a moving platform. Bo et al. [11] used a lever-type quasi-zero stiffness (QZS) vibration isolator to establish a theoretical model with ECD-QZS-VI, further improving the vibration suppression performance in the resonance region. Fulcher et al. [12] and Kashdan et al. [13] studied the structure of buckled beams with bistable and negative stiffness characteristics. They designed a novel Euler beam negative stiffness quasi-zero stiffness (QZS) isolator and established an analytical model for the system and Euler beam. Wang et al. [14] combined two subordinate quasi-zero stiffness mechanisms in parallel with a vertical spring to propose a novel dual quasi-zero stiffness mechanism for a nonlinear ultra-low-frequency vibration isolator. They also analyzed its vibration isolation performance. Sun et al. [6] conducted a study on the use of quasi-zero stiffness structures in vibration measurement. They compared the absolute motion of several vibration platforms with the measured motion signals from these platforms. Xu et al. [15] designed a prototype quasi-zero stiffness (QZS) system that combines a vertical helical spring with two inclined bars connected to magnet springs. The condition of quasi-zero stiffness characteristics can be easily achieved by adjusting the distance between the two magnet springs. Ye et al. [16] designed a quasi-zero stiffness isolation system that supports different loads and can isolate vibrations within the low-frequency range under multiple loads. This system is capable of effectively isolating vibrations in various load scenarios. Zhou et al. [17] and Sun et al. [18] designed a semi-active electromagnetic isolator for tunable hybrid semi-active liquid damper systems (HSLDSs) using magnetic mechanisms. This electromagnetic isolator allows for damping characteristics in the HSLDS system to be adjusted. In the author’s previous study [19], by establishing a six-degree-of-freedom Stewart platform with QZS legs, excellent vibration isolation performance was achieved in six directions.

At present, there are four main ways to convert vibration energy into electrical energy: the electrostatic method [20], a friction motor [21,22,23], electromagnetic method [24,25,26], and piezoelectric method [27,28,29]. Fang et al. [30] drew inspiration from bird motion and designed an isolation system for capturing broadband vibrations and harvesting energy from spacecraft. This system aims to provide wideband vibration isolation capabilities while simultaneously harnessing the generated energy. Yan et al. [31] replaced permanent magnets with springs and designed a four-stable-state piezoelectric vibration energy harvester with geometric nonlinearity in the springs, which improved the power generation of the energy harvester. Wang et al. [32] coupled an electromagnetic generator with a frictional generator to form an energy harvester that collects energy from ultra-low-frequency vibration environments. Additionally, they combined the energy harvester with a quasi-zero stiffness mechanism to enhance the energy harvesting performance. Wang et al. [33] researched extracting energy from low-excitation-level ultra-low-frequency vibrations. They studied a rolling magnet system and proposed a novel quasi-zero stiffness electromagnetic energy harvester (QZS-EMEH). This new design aims to efficiently collect energy from such vibrations. Koszewnik et al. [34] used a cantilever beam with a piezoelectric harvester for structural health monitoring. Lee et al. [35] used mechanical metamaterials and phononic crystals for energy harvesting, which have a wide range of potential applications for a renewable and ecologically benign energy transition. Cheng et al. [36] proposed a novel piezoelectric energy harvesting device with a high density of the energy harvested from highway traffic. Wang et al. [37] improved energy harvesting by repositioning the piezoelectric patch (PZT) in the middle of a fixed–fixed elastic steel sheet instead of the root, as is commonly the case. To reduce the working space of the energy harvesting mechanism, Su and Tseng [38] proposed to design an extended Charpy piezoelectric energy harvester, which increased the output power compared with the traditional energy harvesting system. Yang et al. [39] researched the development of a multi-stage oscillator for ultra-low-frequency vibration isolation and energy harvesting. They addressed the challenges faced by existing oscillators in achieving effective vibration suppression and utilization under ultra-low-frequency excitations. The multi-stage oscillator they proposed aims to overcome these limitations and enable efficient vibration isolation and energy harvesting in such low-frequency scenarios. The theory of nonlocal elasticity and surface elasticity has been used to analyze the nonlinear vibration of nano piezoelectric structures [40,41,42]. Kiani [43,44,45] studied the axial buckled, vibration, and instability of current-bearing bundle elements in nanosystems. So far, most energy harvesting methods are based on amplifying vibrations or combining vibration isolation with energy harvesting. Hsiao and Chung [46], Siahpour et al. [47], and Li et al. [48] used artificial intelligence methods to check the quality of machine-generated questions or sensor fault diagnosis problems. Jia et al. [49] established a high-temperature strain gauge automatic calibration device, which can simultaneously collect the output signals of the high-temperature strain gauge, thermocouple signals, and displacement signals of the grating ruler. The measurement results are used to calculate theoretical mechanical strain. Information data processing is also a very important stage. Good information processing technology can provide us with great help in different areas, such as in using deep nonlinear state space models [50], end-to-end dual stream convolutional neural networks [51], quotient space theory [52], and fusing MG-DTRS and NRS methods [53]. Currently, few scholars have combined quasi-zero stiffness vibration sensing and energy harvesting to study the measurement of absolute motion and the collection of vibrational energy. This paper proposes a quasi-zero stiffness vibration sensing and energy harvesting integration based on a buckled piezoelectric Euler beam. This system allows for the measurement of the absolute displacement of an object without magnifying vibrations. Furthermore, it can convert mechanical energy from vibration into electrical energy to power low-power devices in the vicinity or provide partial electrical energy. This could potentially achieve self-powering integrated quasi-zero stiffness vibration sensing. Combining quasi-zero stiffness vibration sensing with energy harvesting not only allows for the measurement of low-frequency and ultra-low-frequency vibrations of objects but also enables the collection of mechanical energy within the structure. This approach provides a deeper understanding of an object’s vibration characteristics. Moreover, this energy harvesting method can provide partial electrical energy for low-power devices such as active sensors and wireless sensors, reducing the reliance on traditional batteries. This combination of quasi-zero stiffness vibration sensing and energy harvesting offers new insights and methods for the design and optimization of vibration control and energy utilization in structures. It holds significant theoretical and practical implications for improving structural performance, extending device lifespan, and reducing energy consumption. It is expected to bring about more innovations and breakthroughs in engineering practice.

In the author’s previous research [54], a three-axis torsional stiffness (TQZS) was proposed to achieve torsional vibration sensing in three rotational degrees of freedom. The QZS system converts the absolute displacement measurement problem into a relative motion measurement problem by providing a wide-frequency vibration-free point. On one hand, to achieve vibration sensing, it is typically necessary to provide power to the relative motion measurement components through wiring or regular battery replacement. On the other hand, the mechanical energy carried by the vibration itself dissipates into the environment. When the object being measured undergoes small vibrations, the system can realize sensing measurements and collect vibration energy to provide a partial power supply for low-power devices. When the object being measured generates relatively large vibration amplitudes, the system can collect vibration energy to provide green and sustainable power for components such as relative motion measurements and wireless communication. Integrating energy harvesting and vibration displacement sensing will be beneficial for environmental protection and reduce labor and material costs.

Based on the above considerations, quasi-zero stiffness vibration sensing and energy harvesting integration based on a buckled piezoelectric Euler beam is proposed. The piezoelectric Euler beam provides negative stiffness to the entire system, while the vertical spring provides positive stiffness. Due to the quasi-zero stiffness (QZS) structure’s good vibration isolation characteristics in an extensive frequency range, there will be an absolute rest point within the system. When the system is fixed to the object being measured at this rest point, the relative motion measured by the system can represent the absolute motion of the object being measured. On the other hand, as the system vibrates, the piezoelectric ceramic undergoes strain and generates electrical energy, enabling energy harvesting.

## 2. Mathematical Modeling

### 2.1. Structure of the Sensor System

The quasi-zero stiffness vibration sensing and energy harvesting integrated system, as shown in Figure 1, mainly consists of two Euler beams, a vertical spring, and a vibration energy harvesting mechanism. The Euler beam provides negative stiffness to the entire system, while the vertical spring provides positive stiffness. The combination of the Euler beam and the vertical spring forms the quasi-zero stiffness structure. The surface of the Euler beam is equipped with piezoelectric strain patches. When the load mass m undergoes relative reciprocating motion with the object being measured, the Euler beam deforms, and the piezoelectric material undergoes deformation. Due to the piezoelectric effect, charges are generated on the surface of the piezoelectric ceramics, which can power the low-power relative motion sensing component (and wireless communication component) or provide partial electrical energy, thus enabling energy harvesting for absolute vibration measurement.

In the measurement of displacement, the vibration sensing system should be fixed on the measured object. It is assumed that the displacement of the substrate excitation and the load mass are U and S, respectively, for a small relative vibration x=S−U. When the vibration sensor system has the QZS property, the vibration response S should be far less than excitation U. So, U≈−x, that is, the absolute motion can be obtained by measuring the relative motion.

### 2.2. Modeling of the Sensor System

#### 2.2.1. Modeling of the BPEB

Figure 2 shows a schematic diagram of the buckled piezoelectric Euler beam. The ends of the beam are supported on the bracket, and the piezoelectric sheet made of lead zirconate titanate piezoelectric ceramics (PZT-5H) is attached to the middle of the Euler beam. Here, z represents the coordinate perpendicular to the axis, and y represents the coordinate along the axis. One end of the beam is subjected to an axial force F in the y direction, and the displacement in the z direction at the middle of the beam is denoted as w. The output voltage of the piezoelectric material is V.

It is assumed that the deformation and electric field of the piezoelectric ceramic are uniform in the z direction. Under the action of an axial force F, the geometric shape of the simply supported Euler beam can be regarded as a half-sine wave. Then, the deflection at various points on the Euler beam can be expressed as
(1)$$\delta \left(y,t\right)=h\left(t\right)\sin\frac{\pi y}{L}$$
where h is the deflection of the midpoint of the Euler beam, the length of the beam is L, and the strain corresponding to the Euler beam is
(2)$$\varepsilon \left(y,z,t\right)=hz\sin\left(\frac{\pi y}{L}\right)$$

The stress of the Euler beam is
(3)$$\sigma \left(y,z,t\right)=E\varepsilon \left(y,z,t\right)=Ehz{\left(\frac{\pi }{L}\right)}^{2}\sin\frac{\pi y}{L}$$
where E is the Young’s modulus of the Euler beam, and the strain of the piezoelectric ceramic can be approximated to the strain of the beam, and its strain can be expressed as
(4)$$\varepsilon_{t}\left(y,t\right)=\varepsilon \left(y,\frac{h_{L}}{2},t\right)=\frac{h_{L}h}{2}{\left(\frac{\pi }{L}\right)}^{2}\sin\frac{\pi y}{L}$$
where $h_{L}$ is the thickness of the Euler beam. Using the method of small elements, when the element size of the beam is dy and the Euler beam is deformed, the element will rotate by an angle θ, and the projection of the element’s length in the horizontal direction is dycosθ. The displacement l at the end of the beam is
(5)$$l=\int_{0}^{L}\left(1-\cos\theta \right)dy$$

The cosθ value is expanded into a power series, neglecting higher-order terms as follows:(6)$$l=\int_{0}^{L}\frac{\theta^{2}}{2}\left(1-\frac{\theta^{2}}{12}\right)dy$$

The deflection δ and angle θ are related to
(7)$$\theta \approx \frac{\partial \delta }{\partial y}=\cos\frac{\pi y}{L}\cdot \frac{\pi h}{L}$$

Substituting Equation (7) into Equation (6) can be obtained using the following equation:(8)$$h=\frac{L\sqrt{8\left(1-\sqrt{q}\right)}}{\pi }$$
where q=1−l/L. For the piezoelectric cell of this sensing system, the simplified constitutive equation can be written as
(9)$$\left\{\begin{array}{c} \sigma_{t}\left(y,t\right)=Y_{t}\varepsilon_{t}\left(y,t\right)-e_{31}E_{3}\\ u_{3}\left(y,t\right)=e_{31}\varepsilon_{t}\left(y,t\right)+\eta_{33}E_{3} \end{array}\right.$$
where $\sigma_{t}$ is the stress of the piezoelectric material, $Y_{t}$ is Young’s modulus [55], $e_{31}$ is the piezoelectric constant, $E_{3}$ is the electric field strength, $u_{3}$ is the charge areal density, and $\eta_{33}$ is the permittivity. At a uniform electric field strength, the relationship between $E_{3}$ and the output voltage of the piezoelectric material V is
(10)$$E_{3}=\frac{V}{h_{t}}$$
where $h_{t}$ is the thickness of the piezoelectric ceramic [55]. The piezoelectric material output current I is
(11)$$I\left(t\right)=\frac{d}{dt}\int_{D_{t}}u_{3}\left(y,t\right)dD_{t}$$
where $D_{t}$ is the cross-sectional area of the piezoelectric material. The output current of the piezoelectric material can be obtained by substituting Equations (4), (9), and (10) into Equation (11) to obtain Equation (12).
(12)$$I=\frac{\eta_{33}d_{t}L_{t}\dot{V}}{h_{t}}+\frac{\pi e_{31}d_{t}h_{L}\dot{h}\chi }{2L}$$
where $d_{t}$ is the width of the piezoelectric ceramics, $L_{t}$ is the length of the piezoelectric ceramics [55], and $\chi =\left[\cos\left(\pi \left(1-L_{t}/L\right)/2\right)-\cos\left(\pi \left(1+L_{t}/L\right)/2\right)\right]$. Now, substituting Equation (8) into Equation (12) yields the first electromechanical coupling relationship:(13)$$I=\frac{\eta_{33}d_{t}L_{t}\dot{V}}{h_{t}}+\frac{e_{31}d_{t}h_{L}\dot{l}\chi }{2\sqrt{2}L}{\left[q\left(1-\sqrt{\frac{l}{L}}\right)\right]}^{-\frac{1}{2}}$$

Under the virtual displacement λl acting on the Euler beam, the virtual strain energy of the piezoelectric Euler beam is
(14)$$\begin{array}{ll} W_{virtual} & =\int_{V_{t}}\sigma_{L}\cdot \lambda \varepsilon_{t}dV_{t}+\int_{V_{L}}\sigma \cdot \lambda \varepsilon dV_{L}\\ & =\int_{0}^{L}\int_{S_{L}}\sigma \cdot \lambda \varepsilon dS_{L}dy+\int_{\frac{L}{2}-\frac{h_{k}}{2}}^{\frac{L}{2}+\frac{h_{k}}{2}}d_{t}h_{L}\sigma_{t}\lambda \varepsilon_{t}dy \end{array}$$
where $V_{t}$ and $V_{L}$ represent the volume of the piezoelectric layer and the beam, respectively, and $S_{L}$ is the cross-sectional area of the Euler beam, which can be obtained by combining the above equation.
(15)$$W_{virtual}=\frac{\partial h\cdot \lambda l}{\partial l}\left\{\frac{\pi^{4}EI_{L}h}{2L^{3}}\right.+\frac{\pi^{3}hEd_{t}h_{L}{}^{2}}{16L^{3}}\cdot \left[\frac{2\pi L_{t}}{L}+\hat{\chi }\right]-\left.\frac{\pi e_{31}d_{t}h_{L}V\chi }{2L}\right\}$$
where $I_{L}$ represents the cross-sectional moment of inertia of the beam, and $I_{L}$ and $\hat{\chi }$ can be represented by the following equation:(16)$$\left\{\begin{array}{c} I_{L}=\int_{S_{L}}z^{2}dS_{L}\\ \hat{\chi }=\sin\frac{\pi \left(L-l_{T}\right)}{L}-\sin\frac{\pi \left(L+L_{t}\right)}{L} \end{array}\right.$$

The second electromechanical coupling relationship of the piezoelectric Euler beam can be obtained by using the principle of virtual work, which is
(17)$$F=\left\{\frac{\pi^{2}EI_{L}}{L^{2}}\right.+\frac{\pi Ed_{T}h_{t}\sqrt{L-l}h_{L}{}^{2}}{8L^{3/2}}\cdot \left.\left[\frac{2\pi L_{t}}{L}+\hat{\chi }\right]\right\}-\frac{qVe_{31}d_{t}h_{L}\chi }{2\sqrt{2}L}\left(1-\sqrt{q}\right)$$

Extracting the constants from the piezoelectric coupling relationship of two piezoelectric Euler beams can be further simplified as
(18)$$\left\{\begin{array}{c} I=C\dot{V}+\frac{\Gamma \dot{l}}{L^{-3/4}}{\left[\left(L-l\right)\left(L^{1/2}-\sqrt{L-l}\right)\right]}^{-\frac{1}{2}}\\ F=\frac{B}{\sqrt{q}}-\frac{\Gamma V}{L^{-3/4}}{\left[\left(L-l\right)\left(L^{1/2}-\sqrt{L-l}\right)\right]}^{-\frac{1}{2}} \end{array}\right.$$
where Γ is the electromechanical coupling coefficient, C is the internal capacitance, and B is the mechanical parameter, which can be obtained using the following equation:(19)$$\left\{\begin{array}{c} \Gamma =\frac{e_{31}d_{t}h_{L}\chi }{2\sqrt{2}L}\\ C=\frac{\eta_{33}Ld_{t}}{h_{t}}\\ B=\frac{\pi E}{8L^{2}}\left(8\pi I+d_{t}h_{t}h_{L}{}^{2}\right)\left(\frac{2\pi L_{t}}{L}+\hat{\chi }\right) \end{array}\right.$$

#### 2.2.2. Dynamic Modeling of the Integrated System

Assuming that the piezoelectric Euler beam that provides negative stiffness to the sensing system has a projected length D in the horizontal direction, R is the equivalent resistance of the relative motion measuring element, and $C_{v}$ is the damping coefficient. When the mass m moves relative to each other, the Euler beam will deform, and the mass m and the base will produce relative displacement m; the length of the beam is
(20)$$l=L-\sqrt{D^{2}+x^{2}}$$

Substituting Equation (20) into Equation (18) can be carried out to obtain the following:(21)$$F=B\sqrt{\frac{L}{\sqrt{D^{2}+x^{2}}}}-\Gamma V\cdot {\left[\frac{\sqrt{D^{2}+x^{2}}}{L}\left(1-\sqrt{\frac{D^{2}+x^{2}}{L}}\right)\right]}^{-\frac{1}{2}}$$

Under the basic displacement excitation, the equation of motion of the sensing system can be expressed as
(22)$$m\ddot{x}+C_{v}\dot{x}+K_{v}x-NF\sin\beta =-m\ddot{U}$$

In this paper, the number of Euler beams N is taken by two, where β can be represented by the following equation:(23)$$\sin\beta =\frac{x}{\sqrt{\hat{\alpha }}}$$

In the above equation, $\hat{\alpha }=D^{2}+x^{2}$. Under harmonic excitation, the fundamental motion U can be expressed as
(24)$$U=U_{0}\cos\left(\Omega \tau \right)$$

Substituting Equations (21) and (24) into Equation (22) yields the first electromechanical coupling equation of the system as follows:(25)$$m\ddot{x}+C_{v}\dot{x}+K_{v}x-NxB\sqrt{L}{\hat{\alpha }}^{-\frac{3}{4}}+N\Gamma xL^{\frac{3}{4}}V{\hat{\alpha }}^{-\frac{3}{4}}\cdot \left(\sqrt{L}-{\hat{\alpha }}^{\frac{1}{4}}\right)=mU_{0}\Omega^{2}\cos\left(\Omega \tau \right)$$

Assuming that the Euler beams are evenly distributed in space, their output currents I can be described as
(26)$$I=-\frac{V}{NR}$$
where R is the load resistance. Equations (20) and (26) are substituted into Equation (18) to obtain the second electromechanical coupling equation of the system:(27)$$C\dot{V}+\frac{V}{NR}-\Gamma x\dot{x}L^{\frac{3}{4}}\cdot {\hat{\alpha }}^{-\frac{3}{4}}\cdot {\left[\sqrt{L}-{\hat{\alpha }}^{\frac{1}{4}}\right]}^{-\frac{1}{2}}=0$$

Next, let us nondimensionalize the two force–electromagnetic coupling equations.
(28)$$\begin{array}{l} \Omega_{N}=\sqrt{\frac{K_{v}}{m}};\;\omega =\frac{\Omega }{\Omega_{N}};C_{1}=RC\Omega_{N};\;\Lambda =\frac{D}{L};\;\xi =\frac{C_{v}}{2m\Omega_{N}};u_{0}=\frac{U_{0}}{L};X=\frac{x}{L};\\ v=\frac{V}{L\sqrt{K_{v}R\Omega_{N}}};\Psi =\Gamma \sqrt{\frac{R\Omega_{N}}{K_{v}}};\;\hat{\gamma }=\Lambda^{2}+X^{2};\;t=\Omega_{N}\tau ;\;\varphi =\frac{B}{K_{v}L}; \end{array}$$
where $\Omega_{N}$ is the reference frequency, ω is the dimensionless excitation frequency, $C_{1}$ is the dimensionless capacitance, Λ is the dimensionless mechanical parameter, ξ is the damping ratio, $u_{0}$ is the magnitude of the dimensionless excitation [55], X is the dimensionless displacement, v is the dimensionless output voltage, Ψ is the dimensionless electrocoupling coefficient, $\hat{\gamma }$ is also the dimensionless mechanical parameter,t is the dimensionless time, φ is the dimensionless geometric parameter, and the dimensionless parameter is substituted into the two electromechanical coupling equations, which can be simplified to
(29)$$X^{\prime \prime }+2\xi X^{\prime }+X+N\Psi Xv\cdot {\hat{\gamma }}^{-\frac{3}{4}}{\left(1-{\hat{\gamma }}^{\frac{1}{4}}\right)}^{-\frac{1}{2}}-N\varphi X\cdot {\hat{\gamma }}^{-\frac{3}{4}}=u_{0}\omega^{2}\cos\left(\omega t\right)$$
(30)$$v-N\Psi XX^{\prime }{\hat{\gamma }}^{-\frac{3}{4}}{\left(1-{\hat{\gamma }}^{\frac{1}{4}}\right)}^{-\frac{1}{2}}+NC_{1}v^{\prime }=0$$

From Equation (29), we can obtain the dimensionless resilience of the system as
(31)$$F_{re}\left(X\right)=X-N\varphi X{\hat{\gamma }}^{-\frac{3}{4}}$$

The normalized stiffness is
(32)$$K\left(X\right)=1-\frac{N\varphi \left(2\Lambda^{2}-X^{2}\right)}{2{\hat{\gamma }}^{7/4}}$$

To satisfy the condition of quasi-zero stiffness, the stiffness at the static equilibrium point when the system is stationary is zero.
(33)$$\varphi =\frac{\Lambda^{3/2}}{N}$$

To represent the relationship between force and displacement more intuitively and to simplify the calculations, a fifth-order Taylor series expansion is performed around X=0 under small excitation. As shown in Figure 3, the fifth-order Taylor expansion closely approximates the original expression.

When the fifth-order Taylor expansion is adopted, there will be a numerical difference between the approximate value and the real value, but the error of the measurement amplitude range is within 0.31%, which meets the general engineering accuracy requirements. So, the dimensionless electromechanical coupling equation containing irrational fractions is approximated by using the fifth-order Taylor series, and the processed electromechanical coupling equation is
(34)$$\left\{\begin{array}{c} X^{\prime \prime }+2\xi X^{\prime }+\mu_{3}X^{3}+\mu_{5}X^{5}+v\left(\eta_{1}X+\eta_{3}X^{3}+\eta_{5}X^{5}\right)=u_{0}\omega^{2}\cos\left(\omega t\right)\\ v+NC_{1}v^{\prime }-X^{\prime }\left(\eta_{1}X+\eta_{3}X^{3}+\eta_{5}X^{5}\right)=0 \end{array}\right.$$
where
(35)$$\mu_{3}=\frac{3}{4\Lambda^{2}};\;\mu_{5}=-\frac{21}{32\Lambda^{4}};\eta_{1}=\frac{2\Psi }{\sqrt{1-\sqrt{\Lambda }}\Lambda^{3/2}};\;\eta_{3}=\frac{\Psi \left(7\sqrt{\Lambda }-6\right)}{4\Lambda^{7/2}{\left(1-\sqrt{\Lambda }\right)}^{3/2}};\eta_{5}=\frac{3N\Psi \left(35\Lambda -62\sqrt{\Lambda }+28\right)}{128\Lambda^{11/2}{\left(1-\sqrt{\Lambda }\right)}^{5/2}}$$

## 3. Dynamic Response

### 3.1. Performance Analysis

The harmonic balance method is applied to solve the dimensionless force–electric coupling equation of the system and its dynamic response. The excitation term and equation solution of the system are both represented as Fourier series. When the tested object vibrates for one cycle, the piezoelectric beam will experience two identical mechanical states, and the output voltage will also go through two identical cycles. Therefore, the frequency of the output voltage is twice the vibration frequency of the tested object. It is assumed that the relative displacement between the foundation and load mass of the system are in steady-state vibration $X=X_{c}\cos\left(\omega t\right)+X_{s}\sin\left(\omega t\right)$, and the output voltage of the vibration sensing system is $v=v_{c}\cos\left(2\omega t\right)+v_{s}\sin\left(2\omega t\right)$. Considering the case of small excitation, only the main resonance response is studied, and higher harmonic terms are ignored. By equating the left and right sides of the equations involving $\cos\left(\omega t\right)$ and $\sin\left(\omega t\right)$ in the motion equation and equating the left and right sides of the equations involving $\cos\left(2\omega t\right)$ and $\sin\left(2\omega t\right)$ in the voltage equation, the harmonic balance equation is obtained.
(36)$$\left\{\begin{array}{c} \begin{array}{l} 10X_{c}{}^{5}\left(\mu_{5}-\eta_{5}v_{c}\right)+2X_{c}{}^{3}\left(\left.6\mu_{3}+10\mu_{5}X_{s}{}^{2}+15\eta_{5}v_{c}X_{s}{}^{2}+4\eta_{3}v_{c}\right)\right.+25\eta_{5}v_{s}X_{s}X_{c}{}^{4}\\ +4v_{s}X_{s}\left(5\eta_{5}X_{s}{}^{2}+6\mu_{3}\right)X_{c}{}^{2}-16u_{0}\omega^{2}+32\xi \omega X_{s}+5\eta_{5}v_{s}X_{s}{}^{5}+4\eta_{3}v_{s}X_{s}{}^{3}\\ +8\eta_{1}v_{s}X_{s}+\left(16\eta_{1}v_{c}+12\mu_{3}X_{s}{}^{2}+10\mu_{5}X_{s}{}^{4}-16\omega^{2}-5\eta_{5}v_{c}X_{s}{}^{4}\right)X_{c}=0 \end{array}\\ \begin{array}{l} \eta_{5}v_{s}X_{c}{}^{5}+5X_{s}\left(8\mu_{5}-\eta_{5}v_{c}\right)X_{c}{}^{4}+v_{s}\left(15\mu_{5}X_{s}{}^{2}+4\eta_{3}\right)X_{c}{}^{3}+12\eta_{3}X_{s}{}^{3}\\ +X_{s}\left(12\mu_{3}+20\mu_{5}X_{s}{}^{2}+5\eta_{5}v_{c}X_{s}{}^{2}\right)X_{c}{}^{2}+10\mu_{5}X_{s}{}^{5}-16\omega^{2}X_{s}-8\eta_{3}v_{c}X_{s}{}^{3}\\ +\left(15\eta_{5}v_{s}X_{s}{}^{4}+12\eta_{3}v_{s}X_{s}{}^{2}+8\eta_{1}v_{s}-32\xi \omega \right)X_{c}-8\eta_{1}v_{c}X_{s}-112\eta_{5}v_{c}X_{s}{}^{5}=0 \end{array}\\ \left(\eta_{5}X_{s}{}^{4}-16\eta_{3}X_{s}{}^{2}-16\eta_{1}\right)\omega X_{s}X_{c}-16\eta_{3}\omega X_{s}X_{c}{}^{3}-19\eta_{5}\omega X_{s}X_{c}{}^{5}+16v_{c}+128C_{1}v_{s}\omega =0\\ \begin{array}{l} 10\eta_{5}\omega X_{c}{}^{6}+\omega \left(16\eta_{3}-30\eta_{5}X_{s}{}^{6}\right)X_{c}{}^{4}+2\omega \left(8\eta_{1}-5\eta_{5}X_{c}{}^{4}\right)X_{c}{}^{2}+32u_{s}\\ -2\omega \left(8\eta_{1}+4\eta_{3}X_{s}{}^{2}+\eta_{5}X_{s}{}^{4}+128C_{1}v_{c}\right)X_{s}{}^{2}=0 \end{array} \end{array}\right.$$

Solving the system of quaternionic decimal equations for Equation (36) yields the dynamic response of the system, as shown in Figure 4. The dynamic response and output power of quasi-zero stiffness vibration sensing and the energy harvesting integration system based on a buckled piezoelectric Euler beam are analyzed. The default parameters used in the solution process are shown in Table 1.

In general, in the quasi-zero stiffness vibration sensing system, the error is caused by the movement of the load mass m. The measurement accuracy is described by the percentage of the amplitude error ratio $x_{A}$ and the phase difference $u_{P}$ between the excitation u and the measured signal −X. The energy harvesting performance can be described by the electrical power p at the load R.
(37)$$x_{A}=\frac{\sqrt{s_{c}{}^{2}+s_{s}{}^{2}}}{u_{0}}=\frac{\sqrt{{\left(u_{0}+X_{c}\right)}^{2}+X_{s}{}^{2}}}{u_{0}}\times 100\%$$
(38)$$u_{p}=180^{\circ }-\arctan\frac{X_{s}}{X_{c}}$$
(39)$$p=v_{s}{}^{2}+v_{c}{}^{2}$$

To better describe the dynamic sensing characteristics of the system, the following assumptions are made. If $x_{A}\leq 10\%$ and $u_{P}\leq 10{}^{\circ}$, it is considered that relative motion −X can represent absolute motion u. If $x_{A}\leq 5\%$ and $u_{P}\leq 5{}^{\circ}$, the measurement accuracy of the sensing system is considered to be high, otherwise the sensing system is not considered to be within the scope of application of the sensor. Figure 4a shows the percentage of amplitude measurement error $x_{A}$, the measured phase difference $u_{P}$, and the output power p curve. From the percentage of amplitude measurement error curve in Figure 4a, it can be observed that frequency jump phenomena occur between frequencies of 0.2 and 0.33, which may result in larger measurement errors. For ω = 0.35, the percentage of amplitude measurement error is approximately 9%, the measured phase difference is approximately 6.9°, and the output power is $2.8\times 10^{-6}$. At this frequency, it is considered that relative motion −X can represent absolute motion u. For frequencies greater than 0.55, the percentage of amplitude measurement error $x_{A}$ is less than 5%, and the phase difference $u_{P}$ is less than 5°. It can be considered that the sensing system can accurately maintain the phase characteristics between the input and output signals. From the output power curve in Figure 4a, it can be seen that as the frequency increases from 0.55 to 1.8, the output power increases from $5.8\times 10^{-6}$ to $3.3\times 10^{-5}$. The system can provide electrical energy for low-power relative motion sensor components, enabling energy harvesting. It is noted that the negative portion of the derivative on the horizontal axis corresponds exactly to the turning point (unstable solution).

By using the Runge–Kutta method for numerical simulation, the stability of the calculated results is verified. The frequency-amplitude curves obtained by the Harmonic Balance Method (HBM) and the Runge–Kutta Method (RKM) are also shown in Figure 4a. It can be observed from Figure 4a that before the frequency downward jump points, there is a significant difference in the phase difference curve between the HBM curve and RKM. This difference arises from the presence of numerous higher-order harmonics near the frequency jump band. The vibration response calculated by the RKM contains harmonics. When estimating the phase difference from results with numerous harmonics, the error is relatively large. Figure 4b shows excitation frequencies of 0.03, 0.1, 0.2, and 0.3, respectively. It can be observed from Figure 4b that under low-frequency excitation, the displacement response and output voltage contain not only the fundamental frequency component but also higher harmonic components. However, compared to the fundamental frequency component, the amplitude of the higher harmonic components is relatively small. In Figure 4b, as the excitation frequency increases from 0.03 to 0.3, the higher harmonic components gradually disappear, and the vibration response is gradually dominated by the fundamental frequency. It is precisely because the higher harmonic components are significant at low frequencies, and the HBM method ignores these higher harmonic components, that the HBM method and RKM method exhibit certain differences in the low-frequency range. From Figure 4a, it can be seen that except for the points before the frequency of 0.3, the results obtained using the two methods show good consistency.

### 3.2. Influence of Difference Structural Parameters

From the electromechanical coupling shown in Equation (34), it can be seen that the electromechanical coupling coefficient, damping ratio, and measured amplitude parameter changes of the system will affect the performance of the sensing system.

Figure 5 shows the influences of different force–electric coupling coefficients on the percentage of amplitude measurement error, measured phase difference, and output power of the vibration sensing system. It can be observed from the figure that as the force–electric coupling coefficient increases, the range of frequency jump phenomena gradually decreases, and the peak also decreases. At high frequencies, as the force–electric coupling coefficient increases, the percentage of amplitude measurement error and the measured phase difference gradually and slightly increase. When the frequency is 1.5 and the force–electric coupling coefficient increases from 0.06 to 0.12, the percentage of amplitude measurement error increases from 1% to 1.5%. Compared with the low-frequency state, the changes in the force–electric coupling coefficient have little effect on the amplitude ratio and phase difference of the measurement error. At a frequency of 0.8, when the force–electric coupling coefficient increases from 0.06 to 0.12, the percentage of amplitude measurement error is less than 5%, and the measured phase difference is less than 5°. The output power decreases from $1.8\times 10^{-3}$ to $4.9\times 10^{-4}$. At this time, the measuring accuracy of the sensing system is relatively high and can provide some electrical energy for low-power devices. When the frequency is more than 0.4, during the process of increasing the force–electric coupling coefficient from 0.04 to 0.1, the amplitude ratio of the sensing system’s measurement is within 10%, and the measured phase difference is less than 10°. It is considered that relative motion −X can represent absolute motion u. At a frequency of 0.37, during the process of decreasing the force–electric coupling coefficient from 0.12 to 0.06, frequency jump phenomena occurred. Therefore, appropriately increasing the force–electric coupling coefficient can suppress the occurrence of frequency jump phenomena, expand the measurable range, and improve the accuracy and efficiency of the sensing system.

As shown in Figure 6, different damping ratios also have significant impacts on the vibration sensing system. When the frequency is below 0.45, applying a damping ratio greater than or equal to 0.015 will cause frequency jumps, with the percentage of amplitude measurement error exceeding 10%, and the measurement result is inaccurate. When the frequency is above 0.7, as the damping ratio increases from 0.015 to 0.03, the percentage of amplitude measurement error gradually increases but remains below 10%, and the measured phase difference also gradually increases but remains below 10°. At this time, it is considered that relative motion −X can represent absolute motion u. During the process of increasing the damping ratio from 0.015 to 0.03, the peak output power of the sensing system decreases from $3.1\times 10^{-3}$ to $7.9\times 10^{-4}$. It is noted that the negative portion of the derivative on the horizontal axis corresponds exactly to the turning point (unstable solution). In the high-frequency region, the effect of changes in the damping ratio on the output power is small, and appropriately reducing the damping ratio can improve the measurement accuracy of the sensing system. In the low-frequency region, appropriately increasing the damping ratio can suppress frequency jumps. Therefore, it is not advisable to apply a small damping ratio in the measurement process to prevent the amplitude ratio of measurement errors from exceeding 10% when the frequency is below 0.45. However, the damping ratio should not be too large; otherwise, the measurement error in the high-frequency region will increase.

The same as before, the dashed lines in the figure represent the unstable solutions of this system. The motion differential equation of Formula 34 shows that the entire system has two main sources of damping. One comes from the damping of the spring or the damper connected in parallel with the spring. In addition, the energy harvesting mechanism can be equivalently considered as a nonlinear damping to a certain extent. During the vibration process, when the amplitude of relative motion is relatively large, the equivalent damping ratio is also relatively large. Conversely, the equivalent damping ratio is small. Therefore, properly reducing the linear damping and increasing the piezoelectric coupling coefficient can be beneficial for both vibration sensing and vibration energy harvesting.

Generally, it is difficult to simultaneously balance the quasi-zero stiffness system response at both high and low frequencies. An analysis of the changes in the force–electric coupling coefficient and damping ratio on sensing performance reveals that different force–electric coupling coefficients and different damping ratios have slightly different effects on sensing performance. This quasi-zero stiffness vibration sensing system can reduce the peak value of the percentage of amplitude measurement error at low frequencies and the percentage of amplitude measurement error at high frequencies by modifying two parameters: the force–electric coupling coefficient and the damping ratio. As shown in Figure 7, at low frequencies, the peak value of the percentage of amplitude measurement error decreases gradually with the increasing force–electric coupling coefficient. At high frequencies, the percentage of amplitude measurement error increases with the increasing force–electric coupling coefficient, but the impact on the sensing system is relatively small. Similarly, at low frequencies, increasing the damping ratio reduces the percentage of amplitude measurement error. However, unlike the force–electric coupling coefficient, increasing the damping ratio significantly increases the percentage of amplitude measurement error at high frequencies. Combining Equation (34) and Figure 7, it can be seen that the influence of the force–electric coupling coefficient on the dynamic performance of the sensing system is related to the vibration amplitude. When the vibration amplitude is larger, its influence on the dynamic performance of the system is significant, while when the vibration amplitude is smaller, its influence is slight. Therefore, the sensing system proposed in this paper is suitable for adopting a smaller damping coefficient and a larger force–electric coupling coefficient, thereby simultaneously obtaining a wider measurement bandwidth and better high-frequency measurement performance.

From Figure 8, it can be observed that as the measured amplitude increases from 0.03 to 0.1, the curve of the percentage of amplitude measurement error shifts to the right, indicating an expanded range of frequency jump phenomena in the low-frequency range. At a frequency of 0.5, with measured amplitudes of 0.03, 0.05, 0.07, and 0.1, the percentages of amplitude measurement error are 4.1%, 4.1%, 4.5%, and 5.6%, respectively. This indicates that as the measured amplitude increases, the percentage of amplitude measurement error also increases. When the frequency is above 0.5 and the measured amplitudes are below 0.1, and when the percentage of amplitude measurement error is less than 10% and the phase difference is less than 10°, it is considered that relative motion −X can represent absolute motion u. From the phase diagram $u_{P}$, it can be seen that during the process of increasing the measured amplitude from 0.03 to 0.1, the phase difference curve of the measurement and the output power curve shift to the right. As the measured amplitude decreases, the percentage of amplitude measurement error near the resonant peak slightly increases. When the measured amplitude is between 0.03 and 0.1, as the measured amplitude increases, the peak value of the generated output power increases from $4.5\times 10^{-5}$ to $4.1\times 10^{-3}$, which can provide power for surrounding low-power devices or supply partial energy.

### 3.3. Comparison with Different Quasi-Zero Stiffness Vibration Sensing Systems

Figure 9 shows the curves of different quasi-zero stiffness vibration sensing systems after dimensionless transformation. They are, respectively, the buckled piezoelectric Euler beam model curve, the three-spring model [6] curve, and the roller cam model [56] curve. Among these three structural forms, the vertical stiffness is 40 N/m, and the horizontal dimension is 0.4 m. From the figure, it can be seen that when the percentage of amplitude measurement error is 10%, the corresponding frequencies are 0.41, 0.63, and 1.08, respectively. Their corresponding phase differences are 7.5°, 8.6°, and 9.1°, respectively. When the frequency is more than 0.28, the percentage of amplitude measurement error of the buckled Euler beam structure is smaller than that of the three-spring and roller cam structures.

Compared to the three-spring structure and the roller cam structure, the vibration sensor of the buckled Euler beam structure has a larger amplitude error and phase difference in the low-frequency non-measurement range (near the resonance peak) (in fact, the vibration sensor of the buckled Euler beam structure has good vibration energy collection characteristics in this frequency range). However, the initial measurement frequency range is lower, and it has a higher amplitude and phase accuracy in the measurement frequency range (in the high-frequency range). This is mainly due to the vibration sensor of the buckled Euler beam structure using a combination of smaller damping and appropriate force–electric coupling coefficients. In the high-frequency measurement range, the equivalent damping of the force–electric coupling coefficients are very small, which makes the sensor have a wider measurement frequency range and a better measurement performance while collecting vibration energy. This conclusion is consistent with the previous analysis.

### 3.4. Time-Domain Simulation

The following is a time-domain simulation using the fourth- or fifth-order Runge–Kutta method, which considers single-frequency excitation, periodic excitation, and random excitation, respectively, and obtains the time-domain response results of the quasi-zero stiffness piezoelectric Euler beam sensing system under different excitation conditions.

#### 3.4.1. Single-Frequency Excitation

A single-frequency excitation is applied to the QZS vibration sensing and energy harvesting integrated system. Figure 10a–c show the dynamic measurement performance and output voltage over time of the system at different single-frequency excitations with an amplitude of 0.2. It can be seen from the figure that the frequency of the output voltage of the system is twice the frequency of the displacement response, with frequencies of 0.9, 1.4, and 1.9 corresponding to Figure 10a–c As the frequency increases from 0.9 and 1.4 to 1.9, the peak values of the percentage of amplitude measurement error are 9.4%, 5.1%, and 3.5%, respectively, and the peak values of the output voltage are $1.3\times 10^{-2}$, $1.7\times 10^{-2}$, and $2.1\times 10^{-2}$. The peak value of the percentage of amplitude measurement error is less than 10%. Therefore, as the measured frequency increases, the measurement accuracy of the sensor gradually improves, and the output voltage also gradually increases. Figure 10d–f show the dynamic measurement performance and voltage over time of the system at different excitation amplitudes when the frequency is 1.6, with amplitudes of 0.03, 0.06, and 0.09 corresponding to Figure 10d–f. It can be seen from the figure that as the amplitude increases from 0.03 to 0.09, the peak values of the percentage of amplitude measurement error are 3.3%, 3.5%, and 3.6%, respectively, and the peak values of the output voltage are $4.1\times 10^{-4}$, $1.6\times 10^{-3}$, and $3.7\times 10^{-3}$. Therefore, as the measured amplitude increases, the output voltage of the system gradually increases, and the measurement error of the sensing system also increases. When the excitation amplitude is small, the system can provide some electrical energy for surrounding low-power devices. Under small excitation conditions, the measurement accuracy is within 10%, and the measurement accuracy of the sensing system is high. Therefore, the relative motion measured by the system can approximately represent the absolute motion of the measured object.

#### 3.4.2. Periodic Excitation

Figure 11a–d show the time-domain response curves of the system under different periodic excitations. In Figure 11a corresponds to a cycle excitation of $u=0.02\cos\left(1.4t+\pi /3\right)+0.05\cos\left(2.2t-\pi /4\right)$, Figure 11b corresponds to the periodic excitation of $u=0.03\cos\left(2t+2\pi /3\right)+0.1\cos\left(1.1t+3\pi /4\right)$, Figure 11c corresponds to the periodic excitation of $u=0.01\cos\left(1.5t+\pi /2\right)+0.05\cos\left(2t-\pi /6\right)$, and Figure 11d corresponds to the periodic excitation of $u=0.05\cos\left(2.3t+\pi /3\right)+0.04\cos\left(1.3t+\pi /6\right)+0.01\cos\left(1.6t+\pi /4\right)$, respectively. The peaks of the corresponding measured motion are 0.069, 0.089, 0.029, and 0.096; the peak values of the percentage of amplitude measurement error are $2.6\times 10^{-3}$, $6.6\times 10^{-3}$, $3.1\times 10^{-4}$, and $4.3\times 10^{-3}$; and the peak output voltage values are $2.5\times 10^{-3}$, $3.6\times 10^{-3}$, $4.5\times 10^{-4}$, and $4.9\times 10^{-3}$, respectively. Therefore, under different periods of excitation, the system can generate voltage to power low-power devices, and the relative displacement signal measured by the system can approximately represent the absolute displacement signal of the measured object.

#### 3.4.3. Random Excitation

Figure 12 shows the dynamic measurement performance and its frequency spectrum under random excitation. The tested random signal is filtered using a bandpass filter with an upper cutoff frequency of 10 and a lower cutoff frequency of 0.5. Before filtering, the mean of the signal is 0, with a standard deviation of 0.5 in Figure 12a,b, and a standard deviation of 0.2 in Figure 12c,d. The vibration response s values in Figure 12a,c are $8\times 10^{-3}$ and $2\times 10^{-3}$, and the peak values of the output voltage are $4.7\times 10^{-2}$ and $4.4\times 10^{-3}$. The peak values of the vibration response s in Figure 12b,d are $3.4\times 10^{-4}$ and $1.4\times 10^{-3}$. And when the frequency is less than 10, the phase remains near 0. The measurement errors of the measured object’s motion and relative motion, as measured by this system, are within 5%, indicating a high measurement accuracy. The relative motion signal obtained from this system can be used as an approximation of the absolute motion signal of the measured object. The harvested electrical energy can provide a partial power supply for nearby low-power devices.

## 4. Conclusions

An integrated system of quasi-zero stiffness vibration sensing and energy harvesting based on bulked piezoelectric Euler beam is proposed. The following conclusions were obtained:The system utilizes quasi-zero stiffness vibration sensing technology, which enables the measurement of the absolute vibration displacement of the tested object under small excitation. Moreover, the electrical energy harvested by the system can be used to power low-power components or provide partial power, providing an alternative approach for wireless applications.Increasing the electromechanical coupling coefficient can reduce the peak value of the measurement error. The higher the damping ratio, the smaller the peak of output power, the smaller the peak of measurement error in the low-frequency range, and the higher the accuracy of the sensing system. However, at high frequencies, the amplitude of measurement error may increase. A larger amplitude of the tested object results in a higher output power of the system, but it may also decrease the accuracy of the sensing system.The quasi-zero stiffness piezoelectric Euler beam vibration sensing system effectively suppresses frequency jumping phenomena and significantly improves measurement performance in the high-frequency range by using a small damping ratio and a large force–electric coupling coefficient. This flexible parameter adjustment capability allows the system to demonstrate good performance in various operating conditions and applications, resulting in more accurate and reliable measurement results. Compared with the three-spring structure and roller cam structure, the vibration sensor of the Euler beam structure can achieve a wider measurement frequency band and better measurement performance.

Further studies will establish more refined sensor models, such as by incorporating high-order vibration modes of a buckled beam into calculations, to obtain more accurate and realistic sensor dynamic responses. And deep learning and other methods will be applied to optimize the structural parameters of the sensor to achieve better vibration sensing and energy harvesting performance.

## Figures and Tables

**Figure 1 sensors-24-00153-f001:**
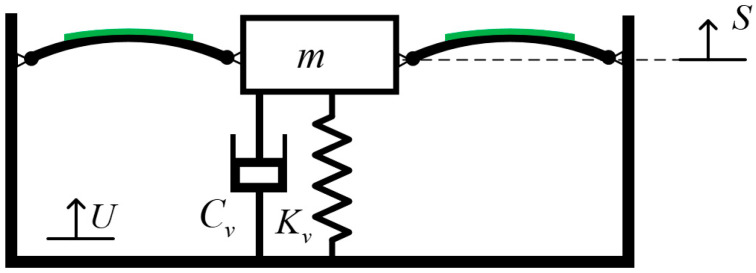
Scheme diagram of the quasi-zero stiffness vibration sensing and energy harvesting integration system.

**Figure 2 sensors-24-00153-f002:**
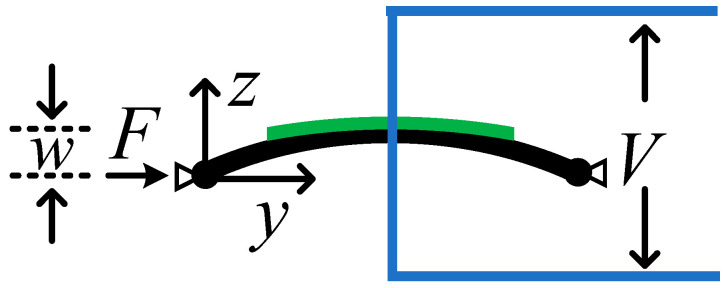
Schematic diagram of buckled piezoelectric Euler beam.

**Figure 3 sensors-24-00153-f003:**
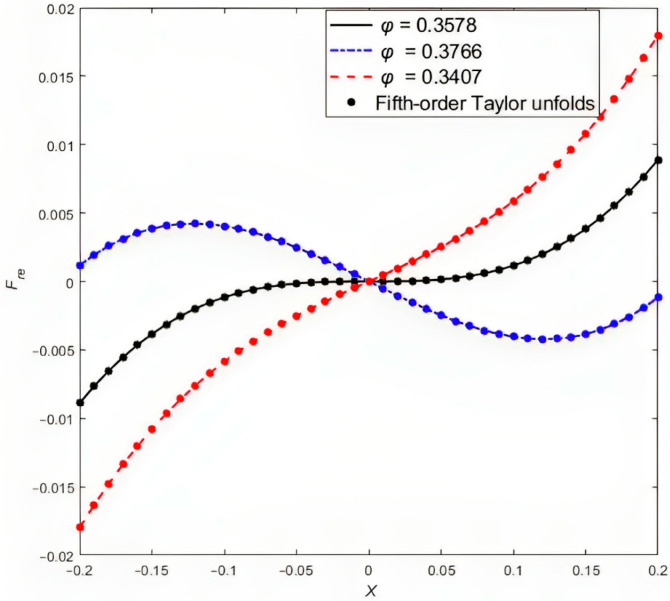
Relationship between force and displacement.

**Figure 4 sensors-24-00153-f004:**
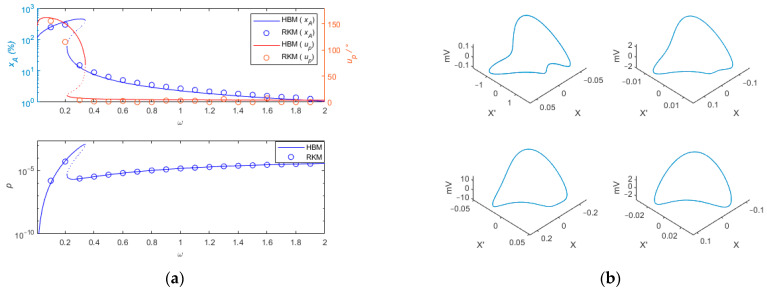
(**a**) shows the frequency domain response plot of the percentage of amplitude measurement error $x_{A}$, measured phase difference $u_{P}$, and output power p calculated using two different methods: the Harmonic Balance Method (HBM) and the Runge–Kutta Method. (**b**) presents the phase plots at excitation frequencies of 0.03, 0.1, 0.2, and 0.3 for the Runge–Kutta Method.

**Figure 5 sensors-24-00153-f005:**
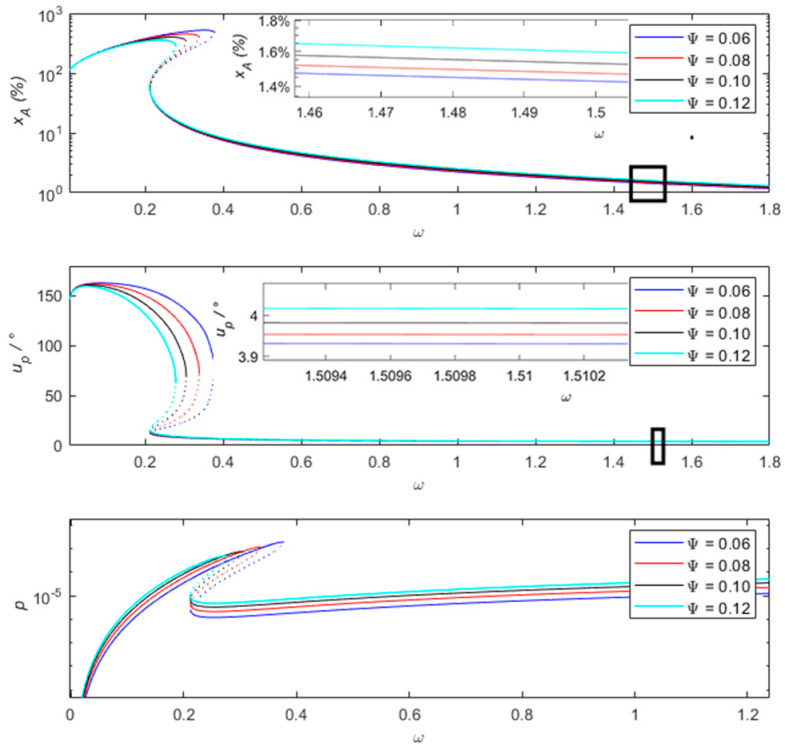
Comparison of the percentage of amplitude measurement error $x_{A}$ of the sensing system and the measured phase difference $u_{P}$ and output power p at different electrocoupling coefficients Ψ.

**Figure 6 sensors-24-00153-f006:**
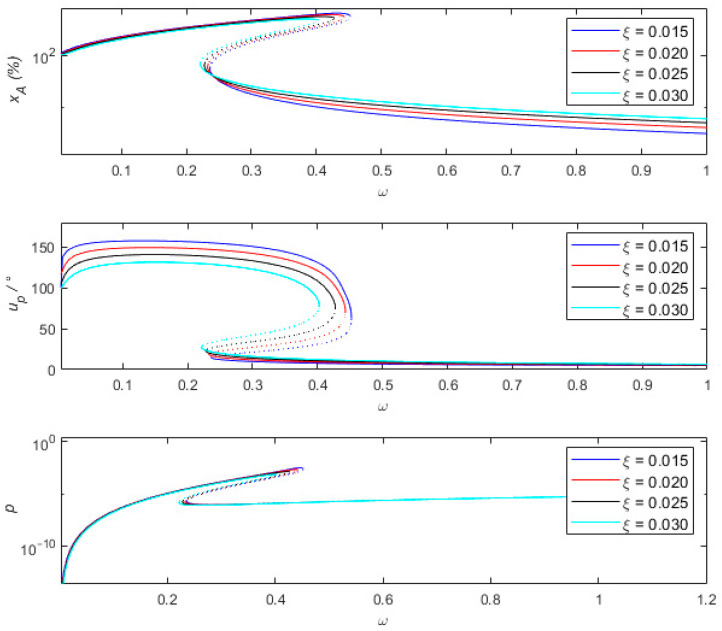
Comparison of the percentage of amplitude measurement error $x_{A}$ of the sensing system and the measured phase difference $u_{P}$ and output power p at different damping ratios ξ.

**Figure 7 sensors-24-00153-f007:**
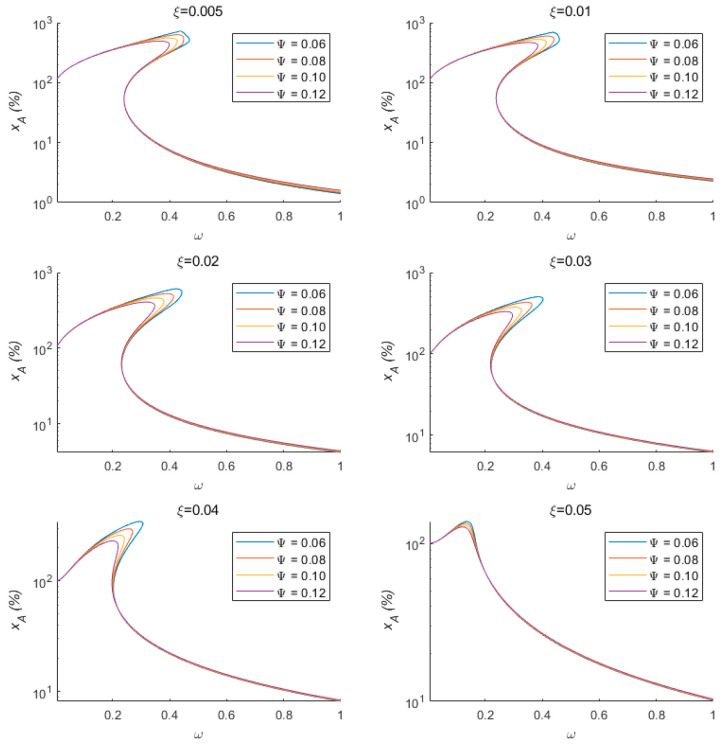
The effects of different force–electric coupling coefficients and damping ratios on sensing performance.

**Figure 8 sensors-24-00153-f008:**
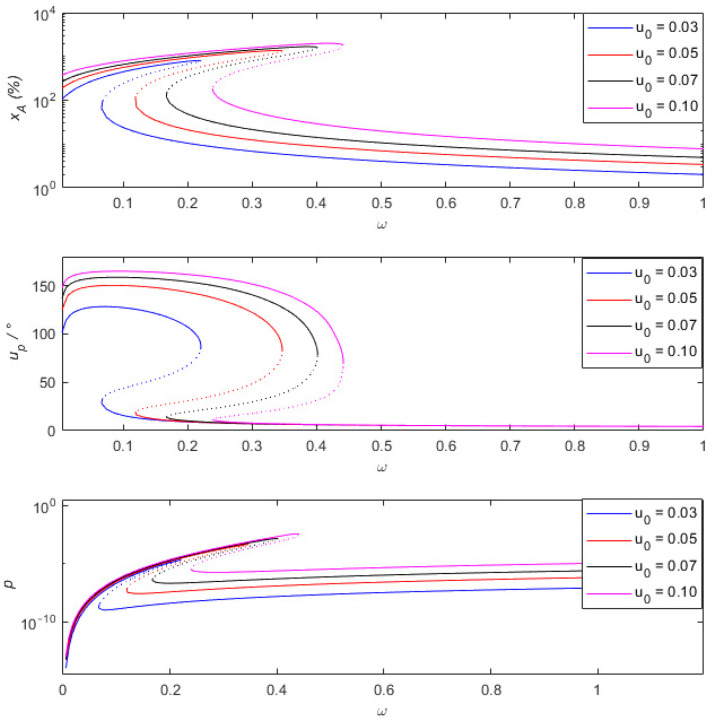
Comparison of the percentage of amplitude measurement error $x_{A}$ of the sensing system and the measured phase difference $u_{P}$ and output power p at different measured amplitudes $u_{0}$.

**Figure 9 sensors-24-00153-f009:**
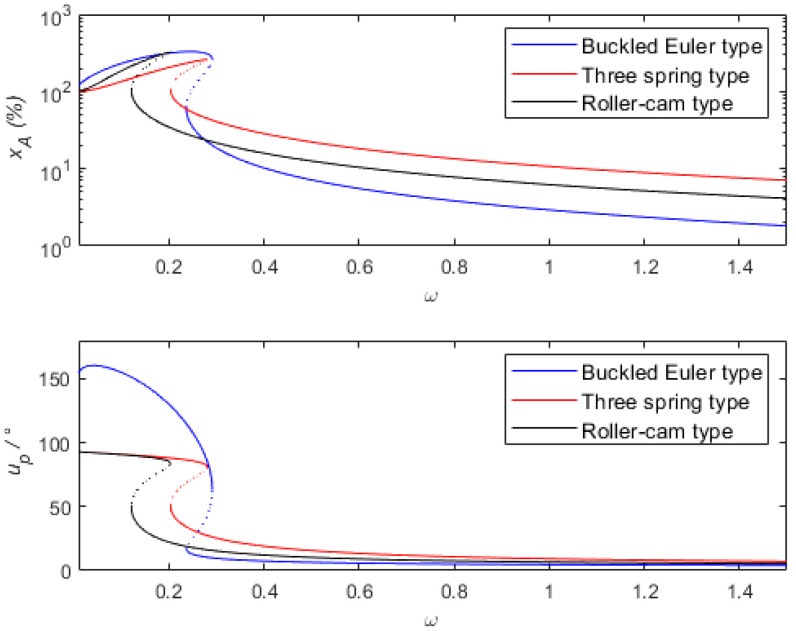
Comparison diagram of vibration sensing systems with different quasi-zero stiffness values.

**Figure 10 sensors-24-00153-f010:**
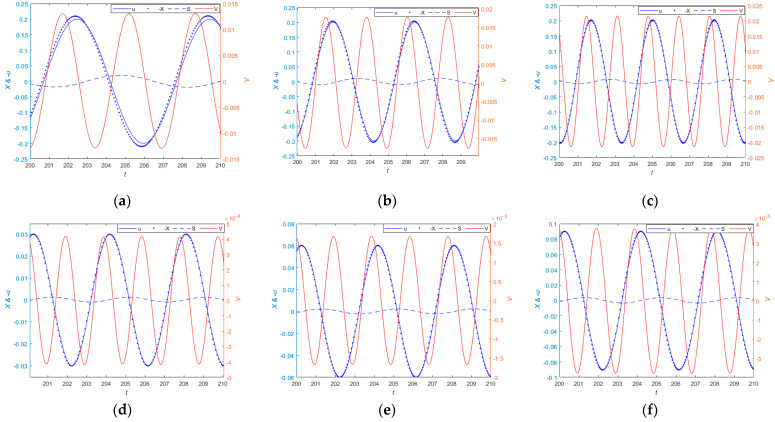
The frequencies corresponding to (**a**–**c**) are 0.9, 1.4, and 1.9, respectively; the amplitudes corresponding to (**d**–**f**) are 0.03, 0.06, and 0.09, respectively.

**Figure 11 sensors-24-00153-f011:**
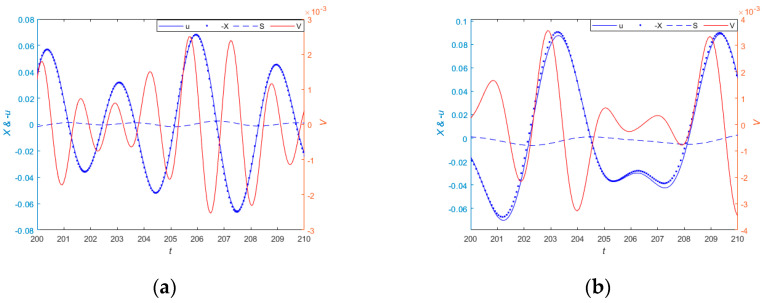
(**a**) shows the measurement performance and the output voltage of the system when the periodic excitation is $u=0.02\cos\left(1.4t+\pi /3\right)+0.05\cos\left(2.2t-\pi /4\right)$. (**b**) shows the measurement performance and the output voltage of the system when the periodic excitation is $u=0.03\cos\left(2t+2\pi /3\right)+0.1\cos\left(1.1t+3\pi /4\right)$. (**c**) shows the measurement performance and the output voltage of the system when the periodic excitation is $u=0.01\cos\left(1.5t+\pi /2\right)+0.05\cos\left(2t-\pi /6\right)$. (**d**) shows the measurement performance and the output voltage of the system when the periodic excitation is $u=0.05\cos\left(2.3t+\pi /3\right)+0.04\cos\left(1.3t+\pi /6\right)+0.01\cos\left(1.6t+\pi /4\right)$.

**Figure 12 sensors-24-00153-f012:**
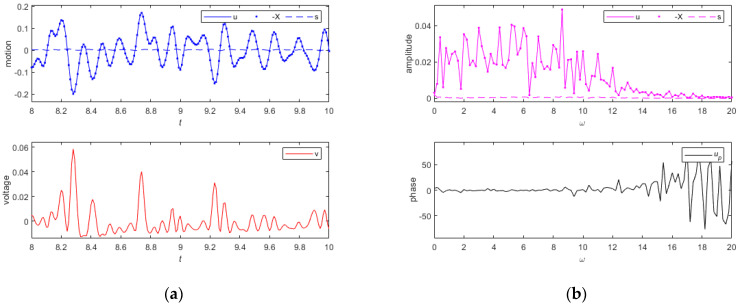
(**a**) The motion and the output voltage diagram of the system are shown when the standard deviation is 0.5. (**b**) The motion amplitude and phase diagram of the system are shown when the standard deviation is 0.5. (**c**) The motion and the output voltage diagram of the system are shown when the standard deviation is 0.2. (**d**) The motion amplitude and phase diagram of the system are shown when the standard deviation is 0.2.

**Table 1 sensors-24-00153-t001:** Quasi-zero stiffness vibration sensing and energy harvesting integration based on buckled piezoelectric Euler beam system structure parameter table.

Symbol	Structural Parameters	Default Value
$K_{v}$	Stiffness of vertical spring	40 N·m^−1^
m	Load object	0.2 Kg
D	Horizontal projection distance of the beam	0.2
R	Equivalent resistance	10^5^ Ω
Λ	Dimensionless mechanical parameters	0.9
$L_{t}$	The length of the piezoelectric ceramic	60 mm
$d_{t}$	The width of piezoelectric ceramics	31 mm
$h_{t}$	The height of piezoelectric ceramics	0.2 mm
$Y_{t}$	Young’s modulus of piezoelectric ceramics	55 GPa
$C_{1}$	Dimensionless capacitance	0.05
ξ	Damping ratio	0.01
$u_{0}$	Dimensionless magnitude	0.1
N	Number of piezoelectric Euler beam	2
Ψ	Dimensionless piezoelectric coupling coefficient	0.08

## Data Availability

The data presented in this study are available upon request from the corresponding author. The data are not publicly available due to privacy reasons.

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
