# Peer review of "Quasi-Zero Stiffness Vibration Sensing and Energy Harvesting Integration Based on Buckled Piezoelectric Euler Beam"

_sensors, 2023, doi:10.3390/s24010153_

Round 1
Reviewer 1 Report
Comments and Suggestions for Authors
Title: Quasi-zero stiffness Vibration Sensing and Energy Harvesting Integration Based on Buckled Piezoelectric Euler Beam
Overall, the manuscript is properly written and well-organized. The paper is focused on the modeling of buckled piezoelectric Euler beam where piezoelectric patches are attached to both Euler beams located in the horizontal direction and next to perform numerical simulations in the time domain as well as performance analysis. In my opinion, process verification of the obtained results should additionally be considered in this manuscript.
Major remarks:
1. Simulations of the BPEB are performed for the chosen resistive load. As it is known, the effectiveness of the piezoelectric vibration-based energy harvesting system is the highest for the optimal resistive load. In my opinion, it should be additionally considered in this paper.
2. What kind of type of piezo-patch authors used in simulations (PZT, PVDF, MFC). Please for comments. Also, please add the parameters of this piezo in Table 1.
3. In the introduction, the authors should consider more references related to the piezoelectric method that were published in the three last years. The instance of these papers are:
- Koszewnik A., Leśniewski K., Pakrashi V., Numerical Analysis and Experimental Verification of Damage Identification Metrics for Smart Beam with MFC Elements to Support Structural Health Monitoring, Sensors 2021, 21(20), 6796; https://doi.org/10.3390/s21206796
4. Signs of load mass m and the resistive load R in text in Section 2 should adjusted to font and size of text. Please also check others signs in text and improve them.
Reviewer 2 Report
Comments and Suggestions for Authors
This paper discusses a unique integration system that combines quasi-zero stiffness vibration sensing and energy harvesting. It is based on a buckled piezoelectric Euler beam (BPEB) with quasi-zero stiffness (QZS) characteristics. The BPEB provides negative stiffness to the system, which creates a vibration-free point within the system, transforming the absolute displacement measurement problem into a relative motion sensing problem. Additionally, during the measurement process, the BPEB collects vibration energy, which can provide electrical energy for low-power relative motion sensing devices. This also helps suppress the frequency range of the jump phenomenon, thereby expanding the frequency domain measurement range of the sensing system. The research shows that this system can measure the absolute motion signal of the tested object in low-frequency vibration with small excitation. By adjusting parameters like the force-electric coupling coefficient and damping ratio, the measurement accuracy of the sensing system can be improved. The system can also convert mechanical energy into electrical energy to power low-power sensors or provide partial power, potentially achieving self-powering integrated quasi-zero stiffness vibration sensing. This offers another approach and possibility for the automation development in wireless sensing systems and the Internet of Things field.
I believe that this paper has some potential for consideration of publication in the Sensors-MDPI journal. However, some major issues need to be addressed before it can be considered for publication. Once all of those issues are reasonably addressed point by point, I can recommend that for publication:
1. The novel aspects of the present investigation are crucial to understanding the significance of this research. In your paper, please explain how the performed study stands out from previous research in this field. By highlighting the unique contributions of your investigations, it is hoped to demonstrate the value of your findings and the importance of continued research in this area.
2. It is important to compare some of the obtained results in this paper with previous works. This will help to strengthen the scientific level and reliability of our present work, making it more suitable for future citations.
3. According to Eq. (1), the assumed deflection field in terms of only “sin(pi*y/L)”, it seems that only the first mode of vibration of the simply supported beam has been taken into account; however, we know that the dynamic deflection of a simply supported beam is expressed as a combination of various modes as follows:
![]()
which is also interpretable based on the separation of variables. As is seen, only the first term of this series has been taken in the analysis. Why?
4. Some appropriate reference works should be given for the material properties given in Table 1.
5. The present work has been conducted in the context of the classical continuum theory, which is commonly used for vibration analysis of macro-beams; however, other advanced theories of elasticity are used for small-scale structures, including nonlocal elasticity theory and surface elasticity theory among others. These have more general formulations compared to the classical governing equations. The authors are highly advised to appropriately explain the above issue in two to three sentences by citing the following original works that deal with the buckling and vibration of beam-like structures at a smaller scale:
· https://doi.org/10.1016/j.physleta.2019.07.006.
· https://doi.org/10.1140/epjp/s13360-020-00144-x.
· https://doi.org/10.1016/j.compstruct.2012.01.023.
· https://doi.org/10.1016/j.jpcs.2016.03.013.
· https://doi.org/10.1142/S0217984915501444.
· https://doi.org/10.1007/s00707-016-1679-1.
The above first three papers deal with vibrations of beam-like piezoelectric structures, while the last three ones explain buckling, vibrations, and instabilities of beam-like elements carrying electrical currents which can be extracted from energy harvesting of the system. All of these works are related to the present investigation conducted by the authors and should be cited and explained suitably in the paper.
6. Could the authors provide some information on the damping mechanism of the studied structure? How this important factor could influence the nonlinear stability of the piezo-electric beam? Please clarify in the paper as well.
7. How the efficiency of the suggested sensing piezo-beam element can be enhanced? The authors are expected to propose some possible suggestions for increasing the accuracy and sensitivity of the system, leading to a more effective sensing structure.
8. Some technical and linguistic errors in the paper should be appropriately revised for the next revision. For the authors’ reference, please pay attention to the following points:
· Line 14: The statement “from system” must be modified to “from the system”.
· Line 19: The statement “as force-electric coupling coefficient” must be modified to “as the force-electric coupling coefficient”.
There are many other grammatical errors that should be improved before the next submission, needing high attention of the authors.
· Throughout the paper: There some long sentences, which make reading the whole paper is somehow hard.
It appears that there are several grammatical errors, typos, and technical issues that require attention from the authors.
I kindly suggest that they review and revise these areas thoroughly. If they can address all of the queries mentioned within the provided guidelines, I would be happy to recommend the paper for publication. This would help expedite the acceptance process for the authors. However, if the authors are unable to provide comprehensive answers and make the necessary revisions, the process of acceptance may take longer or it may be rejected. Therefore, I strongly encourage the authors to address all concerns and make the requested changes to the manuscript, and in return, I would be more than happy to reconsider my decision once these modifications have been fully made.

Some technical and linguistic errors in the paper should be appropriately revised for the next revision. For the authors’ reference, please pay attention to the following points:
· Line 14: The statement “from system” must be modified to “from the system”.
· Line 19: The statement “as force-electric coupling coefficient” must be modified to “as the force-electric coupling coefficient”.
There are many other grammatical errors that should be improved before the next submission, needing high attention of the authors.
· Throughout the paper: There some long sentences, which make reading the whole paper is somehow hard.
It appears that there are several grammatical errors, typos, and technical issues that require attention from the authors.
Reviewer 3 Report
Comments and Suggestions for Authors
Review of Manuscript Sensors-2717096 by Tuo et al.
Submitted to Sensor
Title: Quasi-Zero Stiffness Vibration Sensing and Energy Harvesting Integration Based on Buckled Piezoelectric Euler Beam
General Comments:
This manuscript presents an innovative system for absolute displacement measurements using a buckled piezoelectric Euler beam (BPEB) with quasi-zero stiffness (QZS) characteristics. The BPEB creates a vibration-free point, transforming absolute displacement measurement into relative motion sensing. It also harnesses vibration energy to power low-energy sensors and reduces sudden frequency jumps, thereby broadening the sensing system's frequency domain. This system accurately measures absolute motion during low-frequency vibrations with minimal excitation. Adjusting parameters enhances measurement accuracy, and the system converts mechanical vibration energy into electrical power for low-energy sensors, potentially enabling self-powered quasi-zero stiffness vibration sensing. This innovation offers new possibilities for automation in wireless sensing systems and the Internet of Things field.
According to the high-quality standards of the Sensor, the paper can be considered for publication after some revisions. Some major and minor comments are summarized as follows.
Major/Minor Comments:
l The introduction lacks coherence and fails to provide a comprehensive analysis of the current state-of-the-art in the field. A clearer illustration of the motivation, highlighting deficiencies in existing research, is needed. The paper appears to sequentially present previous studies without elucidating the motivation behind the current work.
l The second and third sections of the paper can be merged to serve as mathematical modeling.
l By utilizing a Taylor expansion and removing higher-order terms, what qualitative and quantitative differences does this simplification bring to the computed results?
l The author employed the harmonic balance method to obtain an approximate solution, followed by presenting a series of parameter analysis results. However, it seems that the vast majority of these outcomes are merely quantitative variations.
l In the analysis of the nonlinear dynamical characteristics, further clarification is needed regarding the stability of the computed results, which the author seems not to have provided. Additionally, by using numerical integration to directly solve the differential equations, the revised paper needs to include a section that compares the approximate solution and the numerical integration solution presented in the paper.
l A single time-history curve may not adequately represent the system's dynamic characteristics, thus additional visuals like phase plots and frequency spectra could be included. Considering that some of the time-history curves do not exhibit periodic solutions, the author needs to further contemplate how to assess these computational results.
l The conclusion section lacks strength and clarity, necessitating an in-depth revision. The excessive wordiness and lack of novelty in some parts of this section undermine its impact.
l The author seems to be able to add some more relevant references to slightly bolster the discussion of existing research.
Comments on the Quality of English Language
Moderate editing of English language required
Round 2
Reviewer 1 Report
Comments and Suggestions for Authors
Thank you for all comments to my remarks and I fully accept them.
As a result, I recommend this paper in present form to publication.
Author Response
Thank you very much for taking the time to review my paper and provide valuable feedback. I am grateful for your guidance and advice, and I am looking forward to the publication of my paper in your journal.
Reviewer 2 Report
Comments and Suggestions for Authors
The comments are in the below file

The previous changes are appreciated and have improved the quality of the English. However, it appears that further modifications are necessary, particularly in terms of technical and grammatical accuracy. To assist the authors, I have provided some guidance below:
- Line 30, the reference numbers “[1][2]” should be modified to “[1,2]”.
- Line 31, the statement “precision engineering[4]and scientific research[5]” should be changed to “precision engineering [4] and scientific research [5]” since we should consider a space between the first bracket and the previous word.
- Line 36, the reference numbers “[6][7]” should be modified to “[6,7]”.
- Line 39, the word “etc” should be removed since it does not give any extra information and its usage is commonly not recommended in scientific articles.
- Line 50&51, the expression “Sun, X. et al. presented Measurement methods for mobile plat-forms[6] [8].” should be modified to “Sun et al. [6,7] presented measurement methods for mobile plat-forms.”, where the reference numbers have been translocated from the end of the sentence to after the corresponding authors’ surnames, and “[8]” has been replaced by “[7]” since the first author of both Refs. [6] and [7] is the same. Please also check that the above-revised expression is acceptable.
- Reference list on pages 22-24, the presented style of some reference works does not successfully meet the journal’s policy, requiring particular attention of authors. In some cases, the full name(s) of the author(s) have been given.
- Line 51, the expression “Jing, X. et al. proposed” should be revised to “Jing et al. [8] proposed” for the sake of more consistency with the reference style format of the journal.
- Line 65, the expression “Kashdan and Fulcher” should be modified to “Kashdan et al. [12] and Fulcher et al. [13]” since both Refs. [12] and [13] have more than three authors and the surname of their first authors should be first given and then “et al.” should be provided.
- Line 98, the statement “Koszewnik A et al.” should be modified to “Koszewnik et al. [35]”.
- Line 99, the statement “Structural Health Monitoring” must be changed to “structural health monitoring”.
- Line 99, the statement “Lee G et al. used” must be modified to “Lee et al. [36]”.
- Line 106, the statement “Su et al. proposed” must be changed to “Su and Tseng [39]” since Ref. [39] only have two authors and their surnames should be connected by “and”.
Therefore, based on the above-mentioned rules, the authors are highly requested that to further revise the present form of the cited references.
- Line 58, the statement “quasi-zero-stiffness isolation system” must be modified to “quasi-zero-stiffness isolation systems” for the purpose of consistency.
- Line 156, the noun “Integration” should be changed to “integration”.
- Line 156, the statement “based on buckled piezoelectric Euler beam” must be revised to “based on a buckled piezoelectric Euler beam”.
- Line 159, the statement “in wide frequency range” must be revised to “in an extensive frequency range”.
- All the given references in the reference list should be also checked and possibly revised in accordance with the journal’s format.
As is seen, there are still many grammatical and style typos as well as unreadable issues that should be appropriately corrected by the authors.
Reviewer 3 Report
Comments and Suggestions for Authors
The authors have satisfyingly answered to almost all the points raised in my initial review.
My opinion is that the paper has been improved, and according to the high quality of SENSORS, the paper can be acceptable after the grammar is double checked.
Comments on the Quality of English LanguageThe authors have satisfyingly answered to almost all the points raised in my initial review.
My opinion is that the paper has been improved, and according to the high quality of SENSORS, the paper can be acceptable after the grammar is double checked.
Author Response
Thank you very much for your valuable feedback on our paper. We have made corresponding changes to the English grammar in this article.
Round 3
Reviewer 2 Report
Comments and Suggestions for Authors
Based on the changes made by the authors in response to the reviewer's feedback, I believe that the second revised version of the article demonstrates significant progress in addressing the previously identified shortcomings. As a result, I believe that this version deserves a high grade of 95/100. I am confident that these improvements have greatly enhanced the quality and relevance of the paper, and I am therefore in favor of its recommendation for publication in Sensors.
Reviewer 3 Report
Comments and Suggestions for Authors
The authors have satisfyingly answered to almost all the points raised in my initial review.
My opinion is that the paper has been improved, and according to the high quality of Sensors, the paper can be acceptable after the grammar is double checked.
Comments on the Quality of English LanguageThe authors have satisfyingly answered to almost all the points raised in my initial review.
My opinion is that the paper has been improved, and according to the high quality of Sensors, the paper can be acceptable after the grammar is double checked.